# Mechanical Properties of Polymethyl Methacrylate as Denture Base Material: Heat-Polymerized vs. 3D-Printed—Systematic Review and Meta-Analysis of In Vitro Studies

**DOI:** 10.3390/biomedicines10102565

**Published:** 2022-10-13

**Authors:** Cláudia Lourinho, Helena Salgado, André Correia, Patrícia Fonseca

**Affiliations:** 1Faculty of Dental Medicine (FMD), Universidade Católica Portuguesa (UCP), 3504-505 Viseu, Portugal; 2Centre of Interdisciplinary Research in Health (CIIS), Faculty of Dental Medicine (FMD), Universidade Católica Portuguesa (UCP), 3504-505 Viseu, Portugal

**Keywords:** polymethyl methacrylate, mechanical tests, denture base, 3D printing, systematic review

## Abstract

The synergy between dentistry and informatics has allowed the emergence of new technologies, specifically 3D printing, which has led to the development of new materials. The aim of this research was to compare the mechanical properties of dental base resins for 3D printing with conventional ones. This systematic review was developed using the PRISMA guidelines, and the electronic literature search was performed with the PubMed/MEDLINE, Web of Science—MEDLINE and EMBASE databases, until 30 April 2022. Two researchers selected the studies independently, and thus eight articles were found eligible for analysis. A meta-analysis was developed to estimate flexural strength. The Cohen’s kappa corresponding to this review was 1.00, and the risk assessment was considered low for the included studies. The 3D printing resin presented lower values of flexural strength and hardness compared with the heat-cured resin. Regarding impact strength, a lower value was recorded for the heat-cured resin compared with the 3D printing resin. Three-dimensional printing resins are viable materials for making prosthetic bases but need further clinical research.

## 1. Introduction

Polymethyl methacrylate (PMMA) is a polymer that has increased in popularity for dental applications due to its unique properties, such as low density, aesthetics, cost-effectiveness, ease of manipulation and adequate physical and mechanical properties. However, prosthetic fractures can occur due to water sorption and poor impact resistance and flexural strength. Consequently, modifications to conventional PMMA have been introduced to improve its properties (such as conductivity, water sorption, solubility, impact resistance, flexural strength and surface hardness) [1,2,3,4,5].

The PMMA used for denture base materials should be biocompatible and should not cause any irritation, toxicity or mutagenicity to oral tissues. Chemically, the PMMA needs to be highly insoluble in saliva and oral fluids. It should be non-reactive with nutrients but should chemically bond to artificial teeth. Thus, the PMMA should have good mechanical properties in order to withstand the forces of mastication without failure [2,3,6]. 

Over the last few years technological changes in dentistry have allowed the use of CAD/CAM technologies with additive manufacturing technique (3D printing). So new materials to be used in 3D printing emerged such as PMMA resins. Therefore, the study of its physical and mechanical properties, as well as its clinical performance, has become fundamental and relevant [7,8,9,10]. 

Accordingly, the aims of this systematic review and meta-analysis were (I) to compare the mechanical properties of conventional PMMA resin with PMMA resin for 3D printing in the manufacture of prosthetic bases, and (II) to synthesize the relevant information on the subject in order to present valuable scientific evidence to adapt the use of available materials to the most appropriate clinical situations.

## 2. Materials and Methods

The systematic review protocol was registered in the PROSPERO—International Prospective Register of Systematic Reviews hosted by the National Institute for Health Research, University of York, Centre for Reviews and Dissemination (code number CRD42022296181) [11].

The study was conducted and reported according to the PRISMA guidelines (Preferred Reporting Items for Systematic Reviews and Meta-Analysis) and PICO approach (Population, Intervention, Comparison and Outcome) [12]. The investigation question was: Does the manufacture of prosthetic bases (P) in PMMA resin for 3D printing (I), compared with conventional PMMA resin (C), present superior mechanical results (O)? 

To answer the PICO question, the inclusion criteria comprised experimental and observational studies, 3D printing PMMA resin, comparative studies with conventional PMMA, studies of mechanical properties and studies written in English. Articles not meeting these criteria were excluded. 

### 2.1. Information Sources and Search Strategy

The electronic literature search was performed by two independent investigators (C.L. and H.S.), covering the period from January 2016 to April 2022 in the PubMed/MEDLINE, Web of Science—MEDLINE and EMBASE databases. An additional literature search of the grey literature was needed. The search keywords were polymethylmethacrylate, 3D printed, flexural strength, impact strength, hardness and a combination of two or more of them.

The search strategy combined different terms: ((“polymethyl methacrylate”[MeSH Terms] OR (“polymethyl”[All Fields] AND “methacrylate”[All Fields]) OR “polymethyl methacrylate”[All Fields] OR (“polymethyl methacrylate”[MeSH Terms] OR (“polymethyl”[All Fields] AND “methacrylate”[All Fields]) OR “polymethyl methacrylate”[All Fields] OR “polymethylmethacrylate”[All Fields] OR “polymethylmethacrylates”[All Fields]) OR (“polymethyl methacrylate”[MeSH Terms] OR (“polymethyl”[All Fields] AND “methacrylate”[All Fields]) OR “polymethyl methacrylate”[All Fields] OR “pmma”[All Fields])) AND (“additive manufacturing”[All Fields] OR (“3D”[All Fields] AND (“printed”[All Fields] OR “printing”[MeSH Terms] OR “printing”[All Fields] OR “print”[All Fields] OR “printings”[All Fields] OR “prints”[All Fields])) OR “printed resin”[All Fields]) AND (“flexural strength”[MeSH Terms] OR (“flexural”[All Fields] AND “strength”[All Fields]) OR “flexural strength”[All Fields] OR ((“impact”[All Fields] OR “impactful”[All Fields] OR “impacting”[All Fields] OR “impacts”[All Fields] OR “tooth, impacted”[MeSH Terms] OR (“tooth”[All Fields] AND “impacted”[All Fields]) OR “impacted tooth”[All Fields] OR “impacted”[All Fields]) AND (“strength”[All Fields] OR “strengths”[All Fields])) OR (“hardness”[MeSH Terms] OR “hardness”[All Fields] OR “hardnesses”[All Fields]))) AND “english”[Language].

### 2.2. Selection of Studies

The review was conducted in three steps. First, two independent reviewers (C.L. and H.S.) evaluated the titles of all the acquired articles according to the inclusion criteria, and discrepancies were resolved by a third reviewer (P.F.). Secondly, the abstracts of the selected titles were screened, and those of interest were marked for full-text analysis. The Cohen’s kappa was determined to evaluate the researchers’ concordance [13].

### 2.3. Data Extraction and Quality Assessment

The selected full-text articles were examined, and the data were tabulated in a standardized Excel software spreadsheet. The following information was extracted from each article: (1) authors’ names and year of publication, (2) brand names and manufacturers, (3) sample size, (4) specimen dimensions, (5) mechanical tests, (6) study objectives and (7) outcome. All data were independently extracted by two reviewers (C.L. and H.S.). 

The criteria used to evaluate the quality of the selected prospective studies were according to the Joanna Brigs Institute Critical Appraisal Checklist for Quasi-Experimental Studies (non-randomized experimental studies), which analyzes the methodological quality of selected studies by answering 9 questions with options of “yes”, “no”, “not clear” or “not applicable”, based on the characteristics of each study [14]. Two independent reviewers (C.L. and H.S.) evaluated the quality of the selected studies, and any disagreement was resolved by a third author (P.F.).

### 2.4. Statistical Analysis

The statistical analysis was performed using CMA 2. In the meta-analysis, the presence of heterogeneity in the data was checked first in order to select the proper model for further analysis. A random effects model was employed to estimate the global effect measure of flexural strength. Forest charts were used to visualize the results with 95% confidence intervals (CI), and the I^2^ index of heterogeneity was also calculated [15].

## 3. Results

The initial electronic search resulted in 93 articles. Titles irrelevant to the research question and duplicate and triplicate titles were excluded. After screening the titles and abstracts, with k = 0.92 and k = 1.00, respectively, 10 articles were qualified for full-text review. Ultimately, eight studies were included for data extraction and analysis—Figure 1. Table 1, Table 2 and Table 3 present a summary of the extracted data.

### 3.1. Study Characteristics

All the studies analyzed were in vitro investigations. Regarding the objectives, all the studies operated with the same purpose, that is, to compare the mechanical characteristics of the prosthetic-based resin for 3D printing with the heat-cured resin. Concerning the mechanical properties studied, seven studies evaluated flexural strength, five impact resistance, three hardness and one elastic modulus and fracture toughness. We also note the existence of studies that evaluated more than one mechanical property.

### 3.2. Quality Assessment

All studies had low risk of bias quality scores according to the Joanna Brigs Institute Critical Appraisal Checklist for Quasi-Experimental Studies (non-randomized experimental studies).

### 3.3. Flexural Strength

Seven of the eight articles under analysis evaluated flexural strength and most recorded higher values for conventional acrylic resin (*n* = 5)—Table 4. 

One of the seven studies that evaluated flexural strength was excluded from the meta-analysis, since it presented data in a graph, which made data extraction impossible. The forest plot represented in Figure 2 reveals that the global mean value comparing the means (effect size= −1.763) of the flexural strengths of the control group and 3D printing group is statistically significant (*p* = 0.02). The variance heterogeneity test indicates a high heterogeneity between the studies: Q = 111.798, *p* = 0.000 and I2 =92% (Figure 3).

### 3.4. Impact Strength 

Five of the eight articles under analysis evaluated impact strength and the results were heterogeneous—Table 5.

### 3.5. Hardness

Regarding hardness, the studies presented results graphically, so they were described in text. Thus, study no. 6 revealed that the hardness of the 3D printing resin presented lower results (30.17 ± 1.38 VHN) compared with the heat-cured resin (41.63 ± 2.03 VHN). It should be noted that the hardness values mentioned above correspond to specimens subjected to thermal cycling in the case of the 3D printing resin and to specimens not subjected to thermal cycling in the case of the heat-cured resin. On the other hand, study nº8 did not present exact hardness values. However, the results allowed us to conclude that hardness was higher in the case of the heat-cured resin compared with the 3D printing resin. The moduli of elasticity and fracture toughness were two properties that were also compared in study no. 5, and although the exact values were not described, the lowest were verified for the 3D printing resin.

## 4. Discussion

The introduction of new technologies in dentistry has allowed the development of new materials, such as 3D printing resins. Therefore, it is essential and relevant to study their general performances and, particularly, their physical and mechanical characteristics, as well as their long-term performances [7,8,9,10,16,17].

The most studied mechanical properties of dental materials are flexural strength, impact strength and hardness. Flexural strength provides an indication of the extent of a material’s resistance to fracture and provides some degree of predictability of its behavior when subjected to static loads, so high values of this mechanical property are clinically relevant for reducing the number of fractures of a prosthetic base. In addition, by subjecting a prosthetic base to the three-point flexion test (the most used test), it is possible to simulate its ability to withstand intraoral functional forces. Effectively, the three-point bending test was adopted by ISO standards as the recommended bending test for polymers, with clinical acceptance and satisfaction for values not lower than 65MPa (ISO 1567:1999) [18,19]. 

Seven studies evaluated this mechanical property, and after comparing the values obtained between the heat-cured resin group (control group) and the 3D printing resin group (study group), it was found that there was greater flexural strength in the case of the control group. Indeed, Al Dwairi et al. 2022 [20] and Prpic’ et al. 2022 [21] justify these results through the internal structure of the materials: the resin of the study group has a lower conversion of monomer into polymer, which can affect the mechanical properties of the material. Furthermore, Perea-Lowery et al. 2021[22] also mention the weak bond between successive layers in 3D printing resins as a justification. Gad et al. 2021 [23] delve into the issue of the connection between successive layers, stating that stratification in a direction parallel to the direction of the load can result in poor adhesion and, consequently, have a negative impact on the resistance of the layer itself. In addition, thermal stress can affect the layering interface, with higher water temperatures increasing water sorption, leading to resin swelling and the separation of the printed layers, which in turn can also affect strength to bending. The same authors found voids at the fracture sites of the printed specimens, so these spaces were identified as factors that contribute to the decrease in mechanical performance. There are two main causes for the creation of voids: the agitation of the resin in the container and in its pouring into the printing vat, and the generation of negative pressure between the resin and air when the printed object’s construction platform is moving. The authors also add that the formation of these voids depends on the viscosity of the printing resin. Thus, the higher the viscosity of the resin, the lower the probability of generating voids. On the other hand, Fiore et al. 2022 [17] demonstrated similar flexural strength values for both groups. Equally, Sonam et al. 2021 [16] also revealed that there were no statistically significant differences between the flexural strength value of the control group and the study group, so they consider that these new materials are a viable option for making prosthetic bases. The similarity of the flexural strength values of both groups can be explained by the resin and the trademarks of the resins used.

Regarding flexural strength, although 3D printing and heat-cured printing are to be standardized (value recommended by ISO 20795-1), there are other materials with better results, from pre-polymerization blocks to milling [17,18,19,20,21,22,23].

The forest plot shown in Figure 2 defines a clear trend in which the resins in the control group show better results in terms of flexural strength. In fact, studies that mention a similarity in flexural strength values for both groups, from a statistical point of view, are not relevant. In any case, these interpretations must be carefully analyzed, since the number of studies is small and there are differences between the resins used, namely, in the resin formulations and the pre- and post-processing stages, and these factors may justify, from a theoretical point of view, these results and this heterogeneity [15,16,17].

Impact strength can have a significant effect on the overall performances of dental prostheses, as it is related to the energy required to fracture a material when subjected to a high-intensity, short-duration force. Impact strength tests are used to assess the amount of energy absorbed by materials before they fracture, usually using the Charpy or Izod methods.

Lee et al. 2022 [24] showed significantly lower impact strength values for the control group compared with the 3D printing group. This fact can be explained by the presence of residual monomers, variations in the powder/liquid ratio, temperature and processing times. Thus, the authors argue that the resin used in the 3D printing group is an alternative to the resin used in the control group. Indeed, the use of 3D printing resins for the manufacture of prosthetic devices contributes to more automated processes, with less risk of introducing errors. According to Sonam et al. 2021 [16], the 3D printing group also showed higher impact strength values compared with the control group. However, in this case, the orientation of the stratification of the layers occurred at an angle of 45° and not parallel to the direction of the load, which may explain these results. On the other hand, Al-Dwairi et al. 2022 [20] found that the differences between the impact resistance values obtained for the control and study groups were not statistically significant. On the other hand, Chhabra et al. 2022 [25] showed higher impact strength values for the control group compared with the 3D printing group, so the difference in the commercial brands of the materials tested may be a possible reason for the results presented in this study, compared with the aforementioned studies. Accordingly, Gad et al.2021 [23] revealed that the impact resistance of the study group was lower than the control group. This fact can be explained by the orientation of the layering when printing the final object, that is, the orientation of the sedimentation of the layers was shown to be parallel to the direction of the impact load [16,20,23,24,25].

Another fundamental mechanical property is hardness, which indicates the extent of resistance of a material to plastic deformation. Thus, the hardness of an acrylic provides an indication of the risk of degradation of the polymer matrix. Therefore, when hardness is reduced, the matrix degrades, increasing the risk of material fracture, as well as the risk of microbial retention and pigmentation. Consequently, the life of the denture base decreases [26]. Hence, Gad et al. 2021 [23] and Prpi´ et al. 2022 [21] produced better results in the control group than in the 3D printing group. These results can be explained by the internal structure of the materials. In addition, the mechanical properties of 3D printing resins are affected by several parameters, such as the orientation of the sedimentation of the layers, the software used, the number and thickness of the layers, the degree of polymerization shrinkage of these same layers and the post-processing steps. 

In view our results, it can be considered that heat-cured resins seem to present better values of flexural strength and hardness. Regarding impact resistance, 3D printing resins appear to perform better. However, the interpretation of these results must be cautious, since there are differences between the commercial brands of the resins used, the compositions and structures of the resins, the pre- and post-processing steps and the software and printers used.

The limitations inherent to this systematic review include a small number of articles, which is justified by the fact that 3D printing resins are relatively recent products whose research studies are still in development. Thus, we consider that an update of this review should be carried out after two years and should aim to also include clinical studies of dentures made by 3D printing technology. Another aspect is that only in vitro studies have been included in this systematic review, where the conditions of the oral environment, such as temperature and masticatory loads, are not reproduced. Therefore, clinical studies are necessary to verify the behavior of these resins in vivo.

## 5. Conclusions

In conclusion, the 3D printing resin showed lower values of flexural strength and hardness compared with the heat-cured resin. Regarding impact resistance, the heat-cured resin obtained lower values compared with the 3D printing resin; 3D printing resins are viable materials for making denture bases. However, to better understand the behavior of these new materials, future studies with strong levels of evidence involving different layering orientations, long-term behavior, the impact of the pre- and post-processing steps and thermocycling tests to simulate the conditions of the intraoral environment are needed. 

Considering that 3D printing resins are relatively recent resins, with little time for clinical follow-up, clinicians should use them with caution, mainly in the fabrication of interim or immediate dentures, as well as in custom tray or record base fabrication for conventional workflow.

## Figures and Tables

**Figure 1 biomedicines-10-02565-f001:**
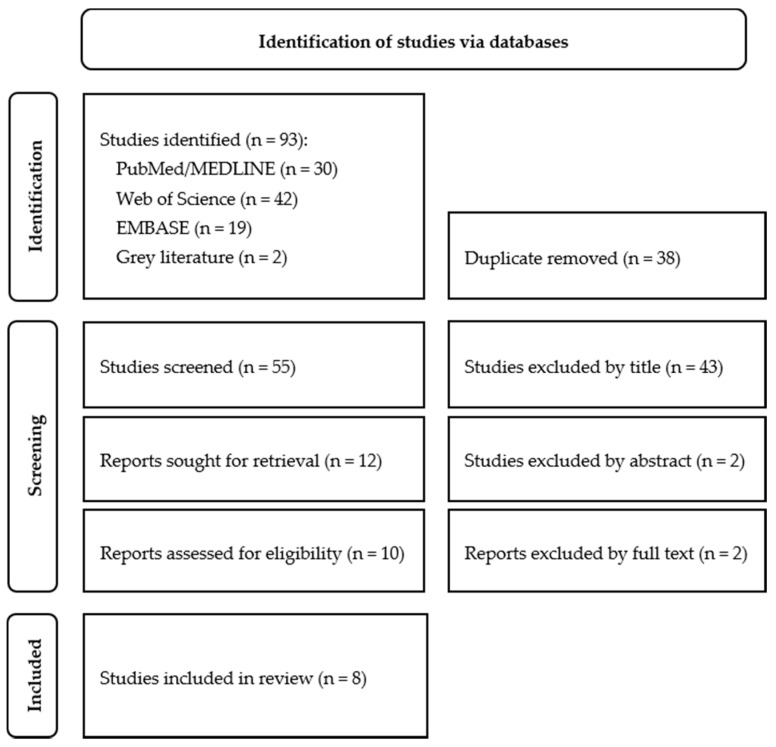
Flow chart of the systematic literature search according to PRISMA guidelines.

**Figure 2 biomedicines-10-02565-f002:**
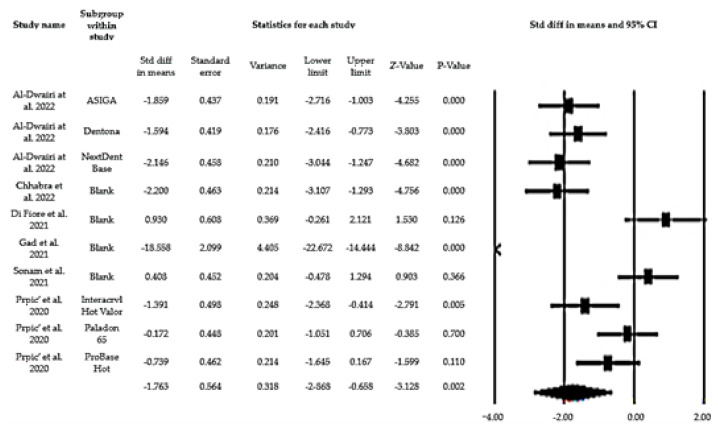
Forest plot for flexural strength.

**Figure 3 biomedicines-10-02565-f003:**
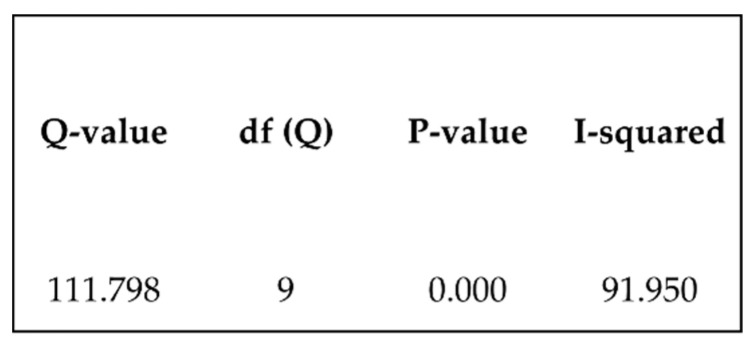
Heterogeneity test.

**Table 1 biomedicines-10-02565-t001:** Identification of studies included in the analysis.

No.	First Author, Year of Publication	Title	Journal	Country
1	Lee J, 2022	Impact strength of 3D printed and conventional heat-cured and cold-cured denture base acrylics	Int. J. Prosthodontics	USA
2	Al-Dwairi ZN, 2022	A Comparison of the Surface and Mechanical Properties of 3D Printable Denture-Base Resin Material and Conventional Polymethylmethacrylate (PMMA)	J. Prosthodont.	Jordan
3	Chhabra M, 2022	Flexural strength and impact strength of heat-cured acrylic and 3D printed denture base resins- A comparative in vitro study	J. Oral Biol. Craniofac. Res.	India
4	Fiore AD, 2022	Comparison of the flexural and surface properties of milled, 3D-printed, and heat polymerized PMMA resins for denture bases: an in vitro study	J. Prosthodont. Res.	Italy
5	Perea-Lowery L, 2021	3D-Printed vs. Heat-Polymerizing and Autopolymerizing	Materials	Finland
6	Gad MM, 2021	Strength and Surface Properties of a 3D-Printed Denture Base Polymer	J. Prosthodont.	Saudi Arabia
7	Sonam D, 2021	Comparative Evaluation of Impact and Flexural Strength of 3D Printed, CAD/CAM Milled and Heat Activated Poylmethyl Methacrylate Resins: An In-Vitro Study	Int. J Sci. Res.	India
8	Prpic’ V, 2020	Comparison of Mechanical Properties of 3D-Printed, CAD/CAM, and Conventional Denture Base Materials	J. Prosthodont.	Croatia

**Table 2 biomedicines-10-02565-t002:** Characteristics of the samples used in the studies analyzed.

No.	Heat-Cured PMMA	3D Printing PMMA	Samples (Number/Size)	Properties	Tests/Machine
1	Lucitone 199 Dentsply Sirona	Denture Base LP Formlabs, USA	50 (25 per group) 64 × 12.7 × 3.2 mm	Impact strength	Izod impact strength test (Monitor Impact Tester, Testing Machines Inc)
2	Meliodent	NextDent, Denture 3D + 3D Systems, USA Dentona 3D, Dentona, Germany DentaBASE, ASIGA, Australia	120 (15 per group) 65 × 10 × 3 mm 60 (15 per group) 25 × 25 × 3 mm	Flexural strength Impact strength Hardness	Three-point bending test, Charpy pendulum test, Vickers hardness test
3	DPI heat cure Dental Products Mumbai, India	Next Dent Denture 3D + 3D Systems, USA	30 (15 per group) 64 × 10 × 3.3 mm 50 × 6 × 4 mm	Flexural strength Impact strength	Three-point bending test Izod impact strength test (International Equipments, India)
4	Aesthetic Blue Clear Candulor	NextDent Denture 3D + 3D Systems, USA	12 (6 per group) 65 × 10 × 3.3 ± 0.2 mm	Flexural strength	Three-point bending test (Universal Testing Machine) (Acumen 3; MTS Systems Corp)
5	Paladon^®^ 65 Kulzer GmbH Mitsui Chemicals Hanau, Germany	IMPRIMO^®^ LC Denture Scheu-Dental GmbH Iserlohn, Germany	48 (16 per group) 10 × 65 × 3.3 ± 0.2 mm	Flexural strength Mod. elasticity Fracture toughness	Three-point bending test (Model LRX; Lloyds Instruments Ltd., Fareham, UK)
6	Major Base.20	NextDent Denture 3D + 3D Systems, USA	60 (12 per group) 64 × 10 × 3.3 ± 0.2 mm 50 × 6 × 4 mm	Flexural strength Impact strength Hardness	Three-point bending test (Universal Testing Machine); Charpy pendulum test, Vickers hardness test and profilometer
7	Not identified	Not identified	40 (10 per group) 64 × 10 × 3.3 mm 50 × 6 × 4 mm	Flexural strength Impact strength	Three-point bending test (Universal Testing Machine) Izod impact strength test (Digital Izod type impact testing machine)
8	ProBase Hot Paladon 65 Interacryl Hot	NextDent Denture 3D + 3D Systems, USA	20 (10 per group) 64 × 10 × 3.3 ± 0.2 mm	Flexural strength Hardness	Three-point bending test; Brinell method

**Table 3 biomedicines-10-02565-t003:** Aims and conclusions of the studies analyzed.

No.	Aim	Conclusions
1	Compare the impact strength of a 3D printing resin with a heat-cured resin.	The 3D printing resin ˃ impact strength than the heat-cured resin.
2	Compare the mechanical properties of three 3D printing resins with a heat-cured resin.	Regarding flexural strength and hardness, the heat-cured resin presented better results. Impact resistance: there were no statistically significant differences between the heat-cured resin and 3D printing resins.
3	Compare the flexural strength and impact strength of a heat-cured resin with a 3D printing resin.	The heat-cured resin presented better results in terms of flexural strength and impact strength compared with the resin for 3D printing.
4	Compare the flexural strength of a heat-cured resin with one for 3D printing.	The heat-cured resin and 3D printing resin showed similar results in terms of flexural strength.
5	Investigate the effects of two post-processing methods on the mechanical properties of a 3D printing resin and compare it with a heat-cured resin.	Post-processing methods impact the flexural strength of 3D printing resins. The resin for 3D printing had inferior mechanical properties when compared with the heat-cured resin.
6	Evaluate the flexural strength, impact strength and hardness of a resin for 3D printing and a heat-cured resin.	The resin for 3D printing had inferior results in flexural strength, impact strength and hardness compared with the thermosetting resin.
7	Evaluate the impact strength and flexural strength of a 3D printing resin and heat-cured resin.	The polymerization process has an influence on impact strength and flexural strength. The 3D printing resin presented higher impact and flexural strength in relation to the heat-cured resin. The impact strength and flexural strength values were higher than the recommended minimum.
8	Evaluate and compare the flexural strength and hardness of different materials and technologies for the manufacture of denture bases.	The resin for 3D printing had lower values of flexural strength and hardness compared with the other group under study.

**Table 4 biomedicines-10-02565-t004:** Average values of flexural strength ^1^.

Nº	Control Group	3D Printing PMMA
2	92.44 ± 7.91	74.89 ± 8.44 (NextDent); 81.33 ± 5.88 (Dentona); 79.33 ± 6.07 (ASIGA)
3	92.01 ± 12.14	69.78 ± 7.54
4	80.79 ± 7.64	87.34 ± 6.39
6	86.63 ± 1.0	69.15 ± 0.88
7	93.90 ± 4.6	95.46 ± 2.84
8	97.35 ± 18.74 (Interacryl Hot) 86.25 ± 20.44 (ProBase Hot) 75.35 ± 18.60 (Paladon 65)	72.25 ± 17.32

^1^ MPa—Megapascal.

**Table 5 biomedicines-10-02565-t005:** Average values of impact strength ^1^.

Nº	Control Group	3D Printing PMMA
1	8.9 ± 0.3	11.2 ± 0.7
2	16.64 ± 1.69	15.20 ± 0.69 (NextDent); 17.98 ± 1.76 (Dentona); 16.76 ± 1.75 (ASIGA)
3	1.67 ± 0.79	1.15 ± 0.40
6	6.32 ± 0.50	2.44 ± 0.31
7	2.08 ± 0.19	3.27 ± 0.12

^1^ Kj/m^2^

## Data Availability

Not applicable.

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
