# Peer review of "Mechanical Properties of Polymethyl Methacrylate as Denture Base Material: Heat-Polymerized vs. 3D-Printed—Systematic Review and Meta-Analysis of In Vitro Studies"

_biomedicines, 2022, doi:10.3390/biomedicines10102565_

Round 1

Reviewer 1 Report

Minor suggestions:

1.     Any particular reason that you focussed only on in vitro studies? Considering you aim at evaluating certain parameters of differently formulated PMMA dentures that are eventually aimed at human applications – I wonder why no in vivo studies have been included? It would be good to have a small section or may be just a small table outlining the most relevant on in vivo studies in this topic.

2.     Please break down the 2nd and 4th paragraphs in discussion into 3 and 2 paragraphs respectively, to make it easier to read and process the conclusions being made.

3.     Please add another section of ‘Limitations’ before or along with the conclusions to reflect upon your work and acknowledge the limitations your current work has. For example your sample size (n=8 studies only) – due to your selection criteria you might have missed a number of studies that could have given you more information relevant to your topic.

Author Response

  1. We appreciate the comment and suggestion. The 3D printing resins for denture’s confection are very recent and there are still few clinical studies to evaluate their performance. Thus, we chose first to evaluate the in vitro mechanical properties of these resins compared to the traditional PMMA. However, we intend to replicate this review in 2 years by also including clinical studies.
  2. We appreciate the suggestion and break down the paragraphs as indicated.

  3. We appreciate the comment and add the limitations of this study before the conclusions.

Reviewer 2 Report

The authors present a comprehensive review of the Properties of Polymethyl Methacrylate as Denture Base Material.

The manuscript is well written, and the study was conducted according to proper methodology.

I propose the following suggestion to improve this already very good manuscript:

The "conclusion" section could be expanded by presenting more clearly the most appropriate clinical situations in which the available materials can best fit.

Author Response

We appreciate the comments and add the information suggested at the conclusion.